# Supervision Incivility and Employee Psychological Safety in the Workplace

**DOI:** 10.3390/ijerph17030840

**Published:** 2020-01-29

**Authors:** Chang-E Liu, Shengxian Yu, Yahui Chen, Wei He

**Affiliations:** 1Mobile E-Business Collaborative Innovation Center of Hunan Province, Key Laboratory of Hunan Province for Mobile Business Intelligence, College of Business Administration, Hunan University of Technology and Business, Changsha 410205, China; liuce15@hutb.edu.cn; 2College of Business Administration, Hunan University of Technology and Business, Changsha 410205, China; yushengxian_cs@citicbank.com; 3School of Business and Tourism Management, Yunnan University, Kunming 650091, China; 4Scott College of Business, Indiana State University, Terre Haute, IN 47809, USA

**Keywords:** employee psychological safety, supervision incivility

## Abstract

Much of the supervision incivility research has focused on the supervisor-subordinate dyad when examining the effects of supervision incivility on employee outcomes. Our study examines a trickle-down effect of supervision incivility across three hierarchical levels, i.e., from the department leader (middle manager), through group leader (supervisor), and to group members (employees), and how it affects group psychological safety. Drawing on a sample of 346 employees and 78 group leaders in 78 work groups, our research found a negative relationship between department leader incivility and group psychological safety, and that this negative relationship was mediated by group leader incivility and moderated by group leader attribution for performance-promotion or injury-initiation motives. We further discuss the theoretical and practical implications of these findings.

## 1. Introduction

Supervision incivility is defined as supervisor’s low intensity deviant behavior with ambiguous intent to harm the subordinate, in violation of workplace norms for mutual respect [1]. Examples of supervision incivility include the superior (leader) publicly criticizes, slanders, or satirizes the subordinate. There is compelling evidence that supervision incivility results in negative employee attitude, behavior, and psychological health [2]. However, it is unclear whether these negative outcomes occur only at the dyadic level between the superior and the subordinate. Meanwhile, Anderson and Pearson [1] pointed out that, in order to define the antecedents of workplace aggression, researchers should investigate the response to mistreatment as a related system of social interactions. In line with this view, our research incorporates supervision incivility as a link in a chain of aggressive workplace events to develop and test a trickle-down effect of supervision incivility. 

The trickle-down effect is an interesting phenomenon in organizations that describes a transfer process of cognition, affection, or behavior from one party to another, such as from leaders to supervisors and then to employees, or from employees to customers [3,4]. Previous research has studied a broad range of organizational topics with regard to the trickle-down effect, such as organizational justice [5], perceived organizational support [6], and psychological contract [7]. Among all the studies on trickle-down effect, leadership research has received the most attention thus far in such aspects as ethical leadership [8], transformational leadership [9], empowering leadership [10], authentic leadership [11], and so on. Obviously, most of the research to date has focused on the trickle-down effect of positive leadership and their impact on organization performance. Fewer studies, however, have examined the trickle-down effect of “dark” leadership behavior [12,13], such as abusive supervision - employees’ perceptions of the extent to which their supervisors engage in sustained hostile verbal and non-verbal behaviors, excluding physical contact [14]. 

Another type of trickle-down effect of dark leadership is supervision incivility, which is lower in intensity than abusive supervision but involves rude or disrespectful behaviors by the supervisors with vague intention to harm their employees [1]. Since workplace incivility is more pervasive than abusive behavior in high-context cultures such as China [15,16,17], and previous research on the trickle-down effect of dark leadership focused on abusive supervision and collected data from low-context cultures such as the United States only [12,13,18], we believe studying the trickle-down effect of supervision incivility in a high-context culture shall be able to complement the existing literature and empirical findings on dark leadership in the workplace. 

To develop our trickle-down model of supervision incivility, we applied the social exchange theory [19]. Social exchange theory posits that individual behavior obeys the rule of reciprocity, which will propel the recipient of the benefit to discharge the obligations by returning the favor, in the hope that continuing such a relationship will bring more valued benefits [20,21]. For example, group leaders who receive fair treatment from their department leaders feel obligated to reciprocate the fair treatment by treating their own subordinates the same way [4]. That is to say, there exists an “indirect exchange” between department leaders, group leaders, and employees. When group leaders experience negative behavior from their department leaders, the norm of reciprocity [21] will lead to a “get even” mentality in the former, which manifests themselves in poor treatment of their subordinates in turn. In other words, a department leader’s uncivil behavior indirectly impacts employees who are two hierarchical levels below the department leader through its effect on their group leader’s uncivil behavior. Thus, we predict that department leaders’ uncivil behavior will be positively related to a group leader’s uncivil behavior, which in turn will be associated with employee attitudes, behaviors, and mental health, as links in a chain of aggressive workplace events. 

As a matter of fact, one aspect of such mental health is the employees’ perceived psychological safety, which is defined as a shared belief among employees as to whether it is safe to engage in interpersonal risk-taking in the workplace [22,23]. Newman et al. [24] further pointed out that psychological safety is likely more potent and meaningful at the group level. Therefore, to specify our exploration of the trickle-down effect of uncivil leadership behavior, we particularly examine whether the impact of department leader incivility on group psychological safety is mediated by group leader incivility.

Last but not least, previous research on abusive supervision has found that not all uncivil behaviors are equal—the subordinate’s perception and attribution of the superior uncivil behavior plays a critical role hereby. In particular, two distinctive types of motives are associated with supervisory abuse—performance promotion and injury initiation [13]. On the one hand, if subordinates perceive their leader’s mistreatment of them as to enhance their performance, the impact of the incivility could be mitigated. On the other hand, if the subordinates believe their leader’s abusive behavior is more likely to harm them on purpose, they would feel harmed more by their leader’s incivility. Therefore, drawing on the attribution literature [25,26], we examine the contingent effect of group leader attribution for performance promotion motive and injury initiation motive [13] on the relationship between department leader incivility and group psychological safety.

In sum, our research tests a mediated-moderation model of supervision incivility and group psychological safety. It can make three unique contributions to the research literature. First, we augment the research on the negative impact of supervision incivility on group psychological safety. Second, we build a trickle-down effect model to unveil how supervision incivility flows top down and consequently undermines group psychological safety. Third, we expand the existing research thoughts on the trickle-down effect of dark leadership behavior by introducing the motive attribution variable as the boundary condition to further clarify its impact on group psychological safety [25]. 

## 2. Theoretical Background

### 2.1. Department Leader Incivility and Group Psychological Safety

As a shared belief by group members with regard to risk-taking in the workplace [27], group psychological safety is formed through the interaction between group members and their leader and is deeply influenced by supervision support [28]. In a psychologically safe work environment, employees feel that their supervisor or colleagues will respect their thoughts and competencies, have positive intentions to them, and will not reject them for saying what they think or engaging in constructive conflict [22,23]. Research to date in this area has found that positive leadership styles, such as transformational leadership [29], ethical leadership [30], change-oriented leadership [31], and shared leadership [32], are all positively and strongly related to group psychological safety. On the other hand, Baumeister et al. [33] found that people are generally more responsive to negative aspects of their external context than to positive ones across a broad range of psychological phenomena, and negative contextual aspects tend to have stronger influences on people’s attitude and behavior than the positive ones. As a type of negative leadership, therefore, supervision incivility may have a greater impact on group psychological safety than positive leadership behavior. 

Meanwhile, a department leader engaging in uncivil behavior such as unfair evaluation, cavil, and contempt may undermine their relationship with group members since the department leader is an indirect leader of the group. The department leader can also make group members feel frustrated, unfair, and vulnerable, and negatively impacts group psychological safety. Moreover, department leader incivility may create a stressful workplace climate, causing group members’ insecurity and anxiety, which also lead to low group psychological safety [34]. Thus, it is reasonable to infer that department leader incivility behavior will weaken group psychological safety.

Furthermore, according to social exchange theory [20,21], interpersonal behavior follows the rule of reciprocity – “you scratch my back and I’ll scratch yours.” One condition for such a reciprocity is that the two parties can achieve their goals by exchanging their unique resources directly [35]. When employees receive favorable treatment from their leader in the workplace, the employees will create value for the organization as a reciprocity responding to their leader [36]. On the contrary, if the department leader engages in negative behavior and treat employees uncivilly, the employees are more likely to exchange negative behavior in return [21]. They might feel unsafe in the work atmosphere and thus tend to withdraw from workplace because they cannot directly retaliate against the department leader. Based on these inferences, we propose our first hypothesis:
**Hypothesis** **1.**Department leader incivility has a negative effect on group psychological safety.

### 2.2. Mediating Effect of Group Leader Incivility

As discussed previously, social exchange theory suggests the norm of reciprocity is the key in regulating social behaviors [20,21]. In other words, benefits received by the group leader from the department leader may be “repaid” by the group leader’s conferring similar benefits on subordinates and not necessarily by repaying the original benefactor (the department leader) directly. This process is called “indirect exchange” by Blau [20], a construct referring to the circuitous chain of exchange in a group caused by normative obligations. Previous empirical research’s findings have supported this argument. For example, Wo et al. [4] found that, when upper-level managers treat supervisors with interpersonal justice and informational justice, supervisors interpret the respectful and dignified treatment and the open and candid communication as strong organizational recognition and support. Then, the norm of reciprocity will drive supervisors to reciprocate the fair treatment by making their own contribution to the organization. One such reciprocal behavior by the supervisors is to treat their own subordinates with respect, dignity, openness, and candidness. In contrast, when supervisors experience unfair interpersonal or informational treatment from upper-level managers, they feel their organization is not supportive. The norm of reciprocity then leads to a “get even” mentality on the part of the supervisors, who are likely to withhold behavior that benefits the organization, such as treating their subordinates nicely. This action by the supervisors in turn lead to low interpersonal and informational justice perceptions among subordinates. In the case of supervision incivility, there might also exist such “indirect exchange.” When group leaders experience incivility from department leaders, the norm of reciprocity might lead to a “get even” mentality on the part of the group leaders and drive them to take similar uncivil actions against their subordinates. Eventually these actions could negatively impact employees’ attitude and behavior, such as their group psychological safety. 

Moreover, when further examining the interactions between the department leader, the group leader, and group members, the group leader stands out in a unique liaison role. Lewin [37] argued that psychologically proximate factors have a more dominant impact on behavior than those less proximate forces. The department leader has frequent, direct contact with the group leader, who in turn directly supervises group members on daily basis. Any uncivil behavior by the department leader will immediately influence the group leader, who then transfers the impact of incivility to his or her direct reports. Specifically, uncivil behavior might serve as the central psychological medium that links department leader incivility to group members’ psychological safety. In other words, the uncivil behavior received by the group leader from the department leader may be “exchanged” to the subordinate, and ultimately undermines group psychological safety. Empirical research lends support to this assumption. For example, Liu et al. [13] found that senior managers’ deviance behavior “trickled down” via its influence on supervisors’ abusive behavior to affect group members’ creativity. Mayer et al. [8] indicated that the behavior of abusive managers trickles down to impact on work group’s interpersonal deviance through abusive supervisor behavior. In addition, Li and Sun [38] reported that a manager’s authoritarian leadership can trickle down to influence employee’s voice behavior through supervisor’s authoritarian leadership. Therefore, we propose:
**Hypothesis** **2.**Group leader incivility mediates the effect of department leader incivility on group psychological safety.

### 2.3. Moderating Effect of Attribution Motive

Research on attribution found that people tend to make causal explanation for the behavior of other individuals around them so as to adjust their own behavior to their social environment [25,26]. In particular, when encountering incivility from their supervisor, subordinates incline to develop causal attribution for the reason or purpose of the supervisor’s incivility as performance-promotion motive or injury-initiation motive [13,14]. Attribution for performance-promotion motive refers to causing injury so as to accomplish an objective such as eliciting high performance, while attribution for injury-initiation motive only refers to causing injury [14]. 

These two types of attribution for supervisor incivility affect subordinates’ behavioral responses by following the norm of reciprocity by the social exchange theory. That is, “benefits” received from one party may be “repaid” by conferring benefits on a third party [39]. Therefore, when a group leader interprets the department leader’s motive as performance promotion, the former might believe that the latter’s incivility is aligned with the former’s own personal interests and will be beneficial to the former’s long-term career development. In other words, the group leader will consider the department leader’s incivility is justified, reasonable, and conducive to the group leader’s own development. Based on social exchange theory, in order to reciprocate the department leader’s kindness, the group leader will implement more incivility on the group members and believe that such uncivil action can further improve the group members’ performance. In contrast, when the group leader interprets the department leader’s motive as injury initiation, the former would perceive the latter’s incivility as unethical and harmful to subordinates’ experiences in their organization. By the same token, in order to resist this kind of unethical and harmful behavior to the subordinates, the group leader will not impose more supervision incivility on the group members. Thus, we propose the following moderating hypothesis:**Hypothesis** **3-1.**Group leader attribution motive for performance promotion on the part of department leader incivility moderates the positive effect of department leader incivility on group leader incivility, such that the positive effect is stronger for group leaders with a high level of attribution for performance-promotion motive.
**Hypothesis** **3-2.**Group leader attribution motive for injury-initiation on the part of department leader incivility moderates the positive effect of department leader incivility on group leader incivility, such that the positive effect is weaker for group leaders with a high level of attribution for injury-initiation motive.

### 2.4. Mediated-Moderation Model

Hypotheses 1 and 2 together propose the mediating effect of group leader incivility and Hypotheses 3-1 and 3-2 propose the moderating effect of attribution for performance-promotion motive and attribution for injury-initiation motive. Following the logic of these two pairs of hypotheses and drawing on Hayes’ [40] recommendation on mediating and moderating effect, we propose a mediated-moderation model:
**Hypothesis** **4-1.**The indirectly negative effect of department leader incivility on group psychological safety via group leader incivility is moderated by attribution for performance-promotion motive, such that the indirect effect will be strengthened for group leaders with high attribution for performance-promotion motive.
**Hypothesis** **4-2.**The indirectly negative effect of department leader incivility on group psychological safety via group leader incivility is moderated by attribution for the injury-initiation motive, such that the indirect effect is weakened for group leaders with high attribution for injury-initiation motive.

Altogether, we summarize our research variables and hypotheses in a conceptual framework in Figure 1. 

## 3. Method

### 3.1. Participants and Procedures

Our research proposal was approved by the academic ethics committees of our institutions. We collected data from 11 business and public service organizations in a southern metropolis of China. In each organization, we distributed our survey questionnaires randomly to group leaders and group members through a liaison from the organization. All the respondents were assured of the anonymity and confidentiality of their responses. We received a total of 346 responses out of 452 copies distributed to group members (response rate 76.5%) and 78 responses out of 108 copies to group leaders (response rate 72.2%). Consistent with previous research [41,42,43,44], we only included groups with five or more respondents. Thus, our final sample included 78 work groups, all of which consisted of five or more employees from the same department in the same organization and their group leaders (supervisors). The final sample consisted of 182 men (42.9%) and 242 women (57.1%), including both group members and their leaders. The survey questionnaire for group members contained measurements on group leader incivility and group psychological safety, while the version for group leaders contained scales measuring department leader incivility, attribution motives, and questions on group size (i.e., number of members) and organization type (i.e., state-owned enterprise and public service, private enterprise, international joint venture enterprise, or wholly foreign-owned enterprise).

### 3.2. Measures

To ensure the reliability and validity of measurements, we adopted well-established scales developed and used by previous researchers in corresponding areas. All the scales were originally developed in English and then translated into Chinese through back-translation validation [45]. They all used five-point Likert scales rated from 1 (strongly disagree) to 5 (strongly agree).

*Department leader incivility and group leader incivility.* We measured supervision incivility with a seven-item workplace incivility scale developed by Cortina et al. [46]. A sample item is “My manager or supervisor put me down or was condescending to me” (α = 0.911). The group leaders were asked to report on the uncivil behavior of their department leader (i.e., middle manager) whereas group members reported on the uncivil behavior of their group leader (i.e., lower managers or supervisors) (α = 0.951). 

*Group psychological safety.* We adopted a seven-term scale developed by Edmondson [27] to measure group psychological safety. A sample item is “If I make a mistake on this team, it is often held against me” (α = 0.836).

*Attribution motives.* We measured attribution motives with a 10-item scale adapted from Liu et al. [13], including five items for attribution for performance-promotion motive (a sample item is “My supervisor desire to elicit high performance from me” (α = 0.750) and five items for attribution for injury-initiation motive (a sample item is “My supervisor desire to cause injury on me” (α = 0.750). 

*Control variables.* To ensure the accuracy and rigor of the results, we controlled group size and organization type since these variables may influence group work results [47].

### 3.3. Data Analyses

We followed a two-step analysis procedure by Anderson and Gerbing [48] to test the hypotheses with SPSS21.0, Mplus7.4, and Hayes’ PROCESS macro [40].

## 4. Results

Table 1 shows the descriptive statistics and correlations among all variables. 

Following Zhou and Long’s [49] suggestions, we first conducted a varimax rotation analysis of principal factors for all variables to examine the presence and magnitude of the common method variance, according to the number of factor precipitation or common factor interpretation. Five common factors (eigenvalue > 1) were extracted from the test results, and the first factor explained only 22.16% of the variance, that is, less than the recommended explanation criterion of 50%. Therefore, we reasonably concluded that the common method variance in the present research was not significant.

Before testing the hypotheses, we examined the distinctiveness of the research variables. We conducted confirmatory factor analyses (CFA) with maximum likelihood estimation in Mplus 7.4 [50]. The CFA results in Table 2 demonstrate that our hypothesized five-factor model (i.e., department leader incivility, group leader incivility, group member psychological safety, attribution for performance-promotion motive, and attribution for injury-initiation motive) was a better fit to the data (x^2^/df = 2.49 < 4, Root-Mean-Square Error of Approximation (RMSEA) = 0.051 < 0.08, Incremental Fit Index (IFI) = 0.909 > 0.9, Tucker-Lewis Index (TLI) = 0.916 > 0.9, Comparative Fit Index (CFI) = 0.931 > 0.9) than these more parsimonious models: a four-factor model (M1) with attribution for performance-promotion motive and attribution for injury-initiation motive loaded on one factor (x^2^/df = 3.92, RMSEA = 0.098, IFI = 0.761, TLI = 0.705, CFI = 0.827); a three-factor model (M2) with group leader incivility, attribution for performance-promotion motive and attribution for injury-initiation motive loaded on one factor (x^2^/df = 4.64, RMSEA = 0.144, IFI = 0.710, TLI = 0.628, CFI = 0.705); a two-factor model (M3) with group member psychological safety, group leader incivility, attribution for performance-promotion motive and attribution for injury-initiation motive loaded on one factor (x^2^/df = 4.98, RMSEA = 0.131, IFI = 0.593, TLI = 0.572, CFI = 0.591); and a one-factor model (M4) with all variables loaded on a single factor (x^2^/df = 5.42, RMSEA = 0.180, IFI = 0.514, TLI= 0.416, CFI = 0.508).

We also generated the group leader incivility and group psychological safety measures by aggregating employee ratings to the group level so as to place all constructs in the model at the work group level of analysis. To justify the aggregation, we assessed the degree of group member consent in terms of group leader incivility and group psychological safety by calculating the r_wg_ statistic [51]. The r_wg_ statistic is used to determine interrater agreement. The median r_wg_ statistic for group leader incivility was 0.76 and the median r_wg_ statistic for group psychological safety was 0.82. Despite considerable dissent on the adequate “cut-off” for r_wg_ value [52], these values are all greater than the generally accepted level of 0.70.

In addition, we computed intra-class correlations (ICCs) to determine the reliability of group leader incivility and group psychological safety [53]. We used ICC (1) to examine the degree of variability in responses at the individual level that is attributed to being part of the group. The ICC (1) was 0.34, F = 5.91, *p* < 0.01 for group leader incivility and 0.32, F = 1.92, *p* < 0.01 for group psychological safety. We used ICC (2) coefficient to examine the reliability of the group means. The ICC (2) was 0.72 for group leader incivility behavior and 0.71 for group psychological safety. These aggregation statistics provided strong support for combining the variables from the individual level to the workgroup level [53].

Next, we used the hierarchical regression analysis method to test Hypothesis 1. As shown in Table 3, we first entered the control variables (group size and organization type) into the regression model and then department leader incivility via stepwise. The results showed a significantly negative correlation between department leader incivility and group psychological safety (M2, β = −0.577, *p* < 0.001). Thus, Hypothesis 1 is supported. 

We tested Hypothesis 2 with Hayes’ PROCESS macro [40], in which 10,000 bias-corrected bootstrapped samples are used. The indirect effect of department leader incivility on group psychological safety via group leader incivility is −0.131, with a 95% confidence interval (CI) [−0.241, −0.068], not including 0 (not shown in Table 3). That is, the indirect effect is significant. Therefore, Hypothesis 2 is supported.

In order to test the moderating role of attribution motives, we first normalized the variable data and used the group-mean-centering technique when testing the interaction effect of department leader incivility and group leader attribution and then conducted regression analysis. As shown in Table 3, department leader incivility had a significant positive correlation to group leader incivility (M3, β = 0.588, *p* < 0.001), after entering the interaction term between department leader incivility and attribution for performance-promotion motive in Model 6, and the interaction coefficient is significant (β = 0.220, *p* < 0.05), R^2^ = 0.39 (*p* < 0.001). Meanwhile, after entering the interaction term between department leader incivility and attribution for injury-initiation motive in Model 7, the interaction coefficient is significant (β = −0.208, *p* < 0.05), R^2^ = 0.351 (*p* < 0.001). These results indicate that attribution for performance-promotion motive played a positive moderating role between department leader incivility and group leader incivility; moreover, attribution for injury-initiation motive played a negative moderating role between department leader incivility and group leader incivility. Therefore, both Hypothesis 3-1 and Hypothesis 3-2 receive support. 

Finally, we tested Hypotheses 4-1 and 4-2 with Hayes’ PROCESS macro [40]. The results showed that, when attribution for performance-promotion and injury-initiation motives are high, the indirect effect of the uncivil manager behavior on the group psychological safety through group leader incivility behavior is −0.272, CI [−0.107, −0.484] and −0.161, CI [−0.355, −0.048], respectively, not including 0; when attribution for performance-promotion and injury-initiation motives are low, the indirect effect of department leader incivility on group psychological safety through group leader incivility behavior is −0.147, CI [−0.042, −0.319] and −0.351, CI [−0.169, −0.624], respectively, not including 0. Together these results suggested that attribution for performance-promotion motive and injury-initiation motive moderate the indirect effect of department leader incivility behavior on group psychological safety through group leader incivility behavior. Nevertheless, according to Hayes [40], if the indirect effects are both significant when the moderation variables are either high or low, the index of moderated-moderation criterion must be employed to determine whether the mediated-moderation effect is significant. In the present research, the index of attribution for performance-promotion motive is −0.062, CI (−0.007, −0.153), while the index of attribution for injury-initiation motive is 0.095, CI (0.229, 0.016), not including 0 for both. Thus, both Hypotheses 4-1 and 4-2 are supported.

Figure 2 and Figure 3 illustrate the moderating effect of group leader attribution motives (i.e., performance-promotion and injury-initiation motives) between department leader incivility and group leader incivility. Figure 2 and slope tests show that department leader incivility was more positively related to group leader incivility (β = 0.110, *p* < 0.05) when group leaders’ attribution for performance promotion motive was high (3.64 + 0.59) than when it was low (β = 0.091, *p* < 0.05)). Figure 3 and slope tests demonstrate that department leader incivility was less positively related to group leader incivility (β = 0.072, *p* < 0.05) when group leader attribution for injury initiation motive was high (1.94 + 0.66) than when it was low (β = 0.089, *p* < 0.05).

## 5. Discussion

We studied the trickle-down effect of supervision incivility in organizations, i.e., how department leader’s incivility affects group leader’s incivility and eventually group members’ psychological safety. Drawing on a sample of 346 employees and 78 group leaders in 78 work groups, we found that department leader incivility has a negative indirect effect on group member psychological safety through group leader incivility and that this trickle-down effect was moderated by group leader attributions. Specifically, group leader attribution of department leader’s incivility for performance promotion motive strengthened the positive influence of department leader incivility on group leader incivility, but group leader attribution of department leader’s incivility for injury initiation motive weakened such influence. These findings are in accord with Liu et al.’s [13] research on the trickle-down effect of executives’ deviance behavior on supervisors’ abusive behavior and Mayer et al.’s [8] study on how supervisors’ abusive behavior trickle down to affect subordinates’ interpersonal deviance in work groups. These findings have meaningful theoretical and managerial implications.

### 5.1. Theoretical Implications

Our findings contribute to the literature on leadership, psychological safety, and attribution in three primary ways. First and foremost, we explored and confirmed the antecedents of supervision incivility in organizations. Previous research on supervision incivility in the workplace has mostly focused on its consequences or dependent variables [2,17] but more or less ignored its antecedents or independent variables, i.e., what could cause supervision incivility. Our research complemented this gap by revealing a trickle-down effect, i.e., the department leader incivility has a positive impact on group leader incivility. That is, the uncivil treatment received by the group leader from the department leader can be transferred to group members through “indirect exchange” according to social exchange theory [20,21]. Therefore, studying the proliferation of supervision incivility from high-rank to low-rank managers expanded the scope of research area on supervision incivility. 

Second, our research enhances the understanding of the role of dark leadership on employee psychological safety. Previous studies on leadership and group dynamics highlighted the impact of positive leadership behavior on group outcomes [54] but paid less attention to the relationship between negative leadership behavior and group outcomes. Even less research has probed the impact of destructive leadership such as abusive supervision and workplace bullying on group outcomes [2,55]. Consequently, the influence of negative leadership on such group outcomes as supervision incivility has generally been left unexplored [2]. We addressed this research gap to provide empirical evidence about the detrimental effect of supervision incivility on subordinate psychological safety. Our research found that department leader incivility behavior has a negative impact on group psychological safety. Department leaders’ uncivil actions could bring in stressful climate in the workplace and eventually threaten group members’ psychological safety. This finding is consistent with other researchers’ results at the individual level [56]. Our research also explores the antecedents of psychological safety. We discovered that department leader incivility and group incivility exerted unique, independent effects on psychological safety. This finding also supports social exchange theory and sheds light on the social exchange nature of workplace incivility [21]. Specifically, if managers are engaged in negative behaviors towards their subordinates, the latter are very likely to respond with negative behaviors in return. Although the subordinates are unlikely to retaliate the leader back directly, they would most likely withdraw from the workplace due to their reduced psychological safety. 

Finally, our research broadened the trickle-down model of leadership by unveiling the contingent roles of subordinate attributions in stimulating or preventing the spread of supervision incivility down the organization hierarchy. Researchers have generally looked at how leaders’ causal attributions for followers’ behaviors may impact leaders’ responses to followers [57,58]. Supervisors respond differently to employee feedback-seeking behavior depending on whether they ascribe employee behavior to performance-enhancement motive or impression management motive [59]. However, empirical research that examines followers’ attributions for leader behavior is still scarce [13]. Answering to Liu et al.’s call [13] about examining followers’ attribution for leader behavior, we showed that followers not only develop two causal motive attributions for leaders’ incivility, but also that such attributions significantly affect the cascading effect of supervision incivility. Thus, the results may generate crucial insights into the research on leadership process and consequences. 

### 5.2. Practical Implications

Our research brings in significant implications to management practice as well. Curbing supervision incivility and minimizing its negative impact on group performance is a direct and effective way to create and maintain a positive and productive work climate for employees and managers to contribute to the success of their organization. Organizations should formulate, publicize, and periodically audit and restate their policies against supervision incivility, destructive leadership, and workplace bullying as part of their equal employment obligation. Organizations might also consider enhancing their managers’ awareness and immunity to supervision incivility through business ethics and regulation compliance training, management development (e.g., on emotional intelligence), individual coaching programs, and so on. Moreover, organizations can also institutionalize their anti-incivility policy through their management systems in terms of managerial selection and promotion, performance appraisal, and rewarding and compensation. On the other hand, as our findings show, supervision incivility from both department leader and group leader can undermine group members’ psychological safety, which is a prerequisite for high group performance and productivity. Organizations, particularly their human resource departments, should take good care of their employees’ psychological safety through such practices as emotional counseling, stress coping and resilience training, and other employee assistance programs. Establishing interactive communication channels between the management and employees such as employee engagement survey, employee opinion e-mailbox, online discussion forum, grievance hotline, or managerial open-door policy can help mitigate employees’ psychological safety as well. 

### 5.3. Limitations and Directions for Future Research

As with any empirical study, our study has several limitations that point to avenues for future research. First, although we draw on social exchange theory [20,21] to explain the trickle-down effect of supervision incivility, alternative theories may be considered to explain the phenomenon, such as social learning theory [12] and affect-driven displaced aggression theory [60]. Further research can deepen the research area by adopting a design that can further test the relative strengths of several alternative theories in explaining the trickle-down effect of supervision incivility. 

Second, we did not empirically test the possible psychological mechanisms between supervision incivility and group member psychological safety, because our research focus on how the conditional, trickle-down effect of supervision incivility from department leader to group leader ultimately damage group member psychological safety. Future study may consider perceived organization support as the mediating process that links supervision incivility and group member psychological safety. 

Third, given the focus of our research model on group leaders’ reactions to department leader incivility, we only investigated group leader attribution motives. However, Martinko, and Gradner [26] argued that both leader and subordinate attributions for subordinate successes and failures might explain variance in leader and subordinate behaviors. Thus, future study may more comprehensively consider the impact of subordinate attribution motivation and leader attribution motivation. 

Last but not least, we collected only cross-sectional data and ignored the time effect on variables. This restrained us from testing the dynamic impact of supervision incivility on group psychological safety, even though our findings suggested that supervisor incivility can be used as a negative predictor of group psychological safety. Future researchers can use a time-series design to collect horizontal and vertical data through empirical sampling or by employing a field test method.

## Figures and Tables

**Figure 1 ijerph-17-00840-f001:**
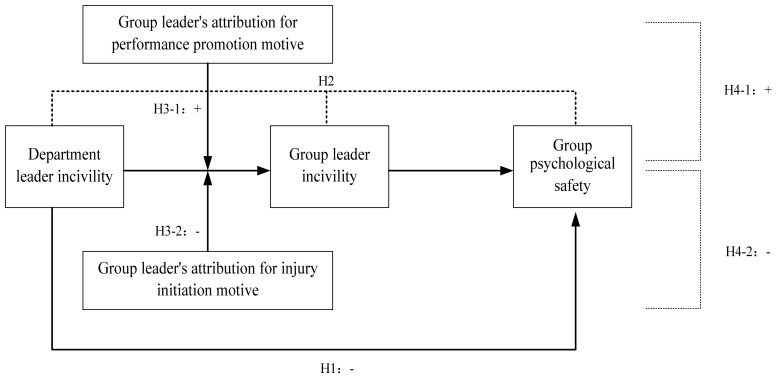
The research conceptual model.

**Figure 2 ijerph-17-00840-f002:**
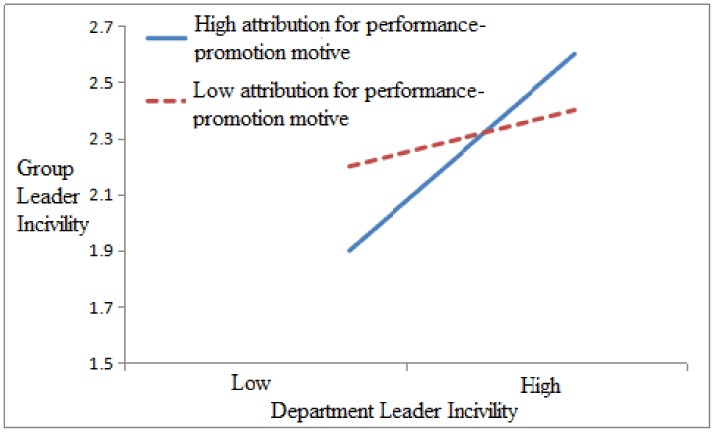
The interactive effect of department leader incivility and group leader attribution for performance promotion motive on group leader incivility.

**Figure 3 ijerph-17-00840-f003:**
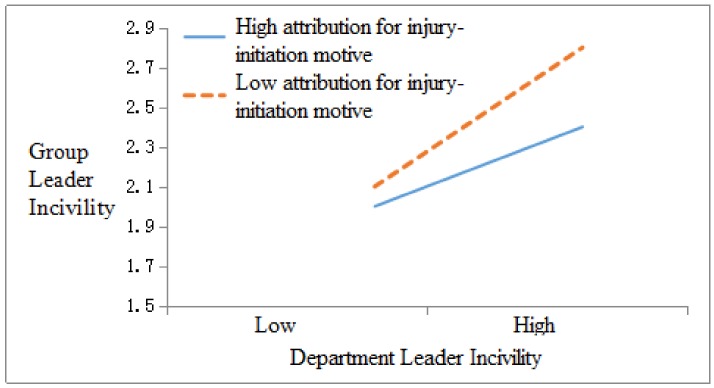
The interactive effect of department leader incivility and group leader attribution for injury initiation motive on group leader incivility.

**Table 1 ijerph-17-00840-t001:** Descriptive statistics of variables (*N* = 424).

Variable	*M*	*SD*	1	2	3	4	5	6	7
1. Size	2.56	0.60	−						
2. Type	1.83	0.63	0.302	−					
3. DLI	2.11	0.70	−0.043	−0.154	(0.911)				
4. GLI	2.13	0.62	0.145	0.064	0.582 **	(0.951)			
5. APPM	3.64	0.59	−0.072	−0.002 *	0.182	0.283 **	(0.750)		
6. AIIM	1.94	0.66	0.036	−0.082	0.310 **	−0.115 *	− 0.411 **	(0.750)	
7. GPS	3.28	0.41	−0.006	0.013	−0.435 **	−0.330 **	0.014 *	−0.101 *	(0.836)

SD: Standard Deviation, DLI = department leader incivility, GLI = group leader incivility, APPM = attribution for performance-promotion motive, AIIM = attribution for injury-initiation motive, GPS = group psychological safety. Reliabilities (Cronbach’s α) are on the diagonal in parentheses. ** *p* < 0.01, * *p* < 0.05.

**Table 2 ijerph-17-00840-t002:** Comparison of measurement models.

Model	x^2^/df	RMSEA	IFI	TLI	CFI
Baseline model	2.49	0.051	0.909	0.916	0.931
M1	3.92	0.098	0.761	0.705	0.827
M2	4.64	0.144	0.710	0.628	0.705
M3	4.98	0.131	0.593	0.572	0.591
M4	5.42	0.180	0.514	0.416	0.508

**Table 3 ijerph-17-00840-t003:** The main and moderating effects (*N* = 424).

Variables		GPS		GLI				
		M1	M2	M3	M4	M5	M6	M7
Control	Size	−0.142	−0.139	0.058	0.079	0.085 *	0.059	0.053
	Type	0.075	−0.015	0.017	−0.678 *	−0.261 *	−0.005	−0.013
Independent	DLI		−0.577 ***	0.588 ***			0.599 ***	0.583 ***
Moderator	APPM				0.346 **		0.251 *	
	AIIM					−0.169 *		−0.102 *
Interaction	IMB × APPM						0.220 *	
	IMB × AIIM							−0.208 *
	*F*	0.73	12.98	12.908	8.160	8.724	11.722	11.404
	*R* ^2^	0.019	0.345 ***	0.344 ***	0.106 *	0.138 *	0.391 ***	0.385 *
	Δ*R*^2^	−0.007	0.326	0.317	0.100	0.132	0.358	0.351

DLI = department leader incivility, TLI = group leader incivility, APPM = attribution for performance-promotion motive, AIIM = attribution for injury-initiation motive, GPS = group psychological safety. *** *p* < 0.001, ** *p* < 0.01, * *p* < 0.05.

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
