# Peer review of "Supervision Incivility and Employee Psychological Safety in the Workplace"

_ijerph, 2020, doi:10.3390/ijerph17030840_

Round 1

Reviewer 1 Report

Thank you for letting me review this article.

The topic is interesting and here are some suggestions relating to this article.

First of all, sample size matters. Please refer to the following link :https://www.frontiersin.org/articles/10.3389/fpsyg.2019.01067/full

In this article, it mentioned "sampling at least 15 level-2 units each in 35 level-3 units results in unbiased fixed effects estimates", however, in your study, only 11 level-3 units were employed.  Please discuss how the sample size would be powerful to support  the results.

Second, the framework of the article seemed a little bit way off. For example, the discussion section didn't relate the findings with other studies. In general, most articles compared findings with previous studied and cited references. In addition, most articles set conclusion/summary before implication/future research. I suggest you to do the same way.  

As for the research framework, the inter-personal level, I think inter-personal psychological safety may better represent individual  psychological safety instead of group psychological safety.

Below are some minor suggestions

Line 56, "the norm of reciprocity"... the claim is strange. Please cite references to support this statement.

Line 65. Please define psychological safety clearer. For example, "being able to show or employ one's self without fear of negative consequences of self-image...."

Author Response

Point 1

First of all, sample size matters. Please refer to the following link: https://www.frontiersin.org/articles/10.3389/fpsyg.2019.01067/full In this article, it mentioned "sampling at least 15 level-2 units each in 35 level-3 units results in unbiased fixed effects estimates", however, in your study, only 11 level-3 units were employed. Please discuss how the sample size would be powerful to support the results.

Response 1

We agree with you on the importance of sample size and thank you for recommending such an important and latest article on this issue. It enabled us to update our knowledge on sample size. As we described in our manuscript (l. 213, p. 5, revised version), however, “11” is the number of organizations where our data were collected, not that of the departments (level-3). We reviewed our database and found that the number of departments in the 11 organizations involved in our research is 43 (level-3), which is not mentioned in our manuscript since the focus of our study is the group level (level-2).

Point 2

Second, the framework of the article seemed a little bit way off. For example, the discussion section didn't relate the findings with other studies. In general, most articles compared findings with previous studied and cited references. In addition, most articles set conclusion/summary before implication/future research. I suggest you to do the same way. 

Response 2

We appreciated this helpful advice. Accordingly we merged the conclusions and the discussion parts and enhanced our discussion by associating our findings with more references. We also would like to point out that our original manuscript has actually referred to or compared with certain previous studies in the “theoretical implications” section, such as references [2], [13], [17], [56], etc. (pp. 10-11, revised version). 

Point 3

As for the research framework, the inter-personal level, I think inter-personal psychological safety may better represent individual psychological safety instead of group psychological safety

Response 3

Our research focuses on group psychological safety, which is defined as a shared belief among employees as to whether it is safe to engage in interpersonal risk-taking in the workplace [22-23]. Newman et al. [24] further pointed out that psychological safety is likely more potent and meaningful at the group level than at the individual level. Moreover, we felt that “interpersonal psychological safety” seemed to include at least two types of dyad psychological relationship – those between a group member and the supervisor, and those among group members. Thus a more general concept of “group psychological safety” should be able to cover both types of interpersonal psychological safety.

Point 4: Line 56, "the norm of reciprocity"... the claim is strange. Please cite references to support this statement.

Response 4

Thanks for pointing out our casual error. We added an existing reference [21] to it in the manuscript.

Point 5

Please define psychological safety clearer. For example, "being able to show or employ one's self without fear of negative consequences of self-image...."

Response 5

We accepted your suggestion and added the following sentence to further elaborate the concept of psychological safety in line 91-94 (revised version): “In a psychologically safe work environment, employees feel that their supervisor or colleagues will respect their thoughts and competencies, have positive intentions to them, and will not reject them for saying what they think or engaging in constructive conflict [22-23].”

Reviewer 2 Report

The topic of research is very interesting, few researchers focus on the effects of this type of leadership behavior and so clearly.

Highlight, the few data offered by the authors on the characteristics of the participants, not only the type of public or private company.

In this regard, It would be interesting to know in more detail sociodemographic characteristics of the participants, type of company and work characteristics. The context studied include health workers? social workers? or teachers? What were their job characteristics (for example years of experience in the company, level of studies, ..).

Did the authors study any of these variables? If not, do you think it is a study limitation?

Author Response

Point 1 Highlight, the few data offered by the authors on the characteristics of the participants, not only the type of public or private company. In this regard, It would be interesting to know in more detail sociodemographic characteristics of the participants, type of company and work characteristics. The context studied include health workers? social workers? or teachers? What were their job characteristics (for example years of experience in the company, level of studies, ..).Did the authors study any of these variables? If not, do you think it is a study limitation?

Response 1

Our research studies the trickle-down effect of supervision incivility from the leader through the supervisor to the group members. Therefore our focus is group behavior, such as group psychological safety, rather than individual behavior. That is why we did not pay much attention to the individual characteristics of our respondents. Yes, our sample did include some teachers. Of course future research can concentrate more on the impact of supervision incivility on individual employees and highlight the individual characteristics.